# Two New Species of Ripipterygidae (Orthoptera, Tridactyloidea) from Mid-Cretaceous of Myanmar with a Key to the Genera of Tridactyloidea in Amber [note 1]

**DOI:** 10.3390/insects13110979

**Published:** 2022-10-25

**Authors:** Jun-Jie Gu, Cheng-Jie Zheng, Dong Ren, Cheng-Quan Cao, Yan-Li Yue

**Affiliations:** 1College of Agronomy, Sichuan Agricultural University, Chengdu 611130, China; 2College of Life Sciences, Capital Normal University, 105 Xisanhuanbeilu, Beijing 100048, China; 3College of Life Sciences, Leshan Normal University, Leshan 614004, China

**Keywords:** mud crickets, *Magnidactylus*, revision, Kachin Province, north Myanmar

## Abstract

**Simple Summary:**

*Magnidactylus* Xu, Fang and Jarzembowski (2020) revised and transferred Ripipterygidae (Tridactyloidea). Two new species of the genus from north Myanmar amber are described. These new findings increase the diversity of fossil Tridactyloidea and provide new knowledge regarding itsmorphology in mud crickets. Additionally, a key to the genera of ambers in Tridactyloidea is provided.

**Abstract:**

The abundance of insects in Burmese amber illustrates a highly diverse orthoptera community of the mid-Cretaceous, but the records of ripipterygids are relatively rare. Here, we reviewed the genus of *Magnidactylus* (Xu, Fang and Jarzembowski, 2020) and transfered it from Tridactylidae to Ripipterygidae. Based on four ambers specimens collected from northern Myanmar, two new species, *Magnidactylus*
*mirus* sp. nov. and *Magnidactylus*
*gracilis* sp. nov., wereerected. *M.*
*mirus* sp. nov. can be characterized by its basal segment and apical segment of paraproctal lobes, which are equally thick and clavate. *M.*
*gracilis* sp. nov. can be characterized by its apical segment of paraproctal lobes, which are distinctly swollen. Additionally, in order to facilitate the classification of amber specimens of Tridactyloidea, a key to the genera ofambers in this superfamily is provided.

## 1. Introduction

Tridactyloidea is an ancient group belonging to Caelifera and has the following distinctive morphological characters: Pro- and meso-tarsi with only two tarsomeres; metatarsi reduced to a single tarsomere; and the paraproct bearing distinctive cerciform lobes (Cylindrachetidae without paraproctal lobes) [1,2]. Tridactyloidea comprises three families: Tridactylidae, Cylindrachetidae and Ripipterygidae. Ripipterygidae are characterized by unsegmented cerci (the cerci are two-segmented in Tridactylidae), uninflated mesotibiae (the mesotibiae are distinctly inflated in Tridactylidae) and modified paraproctal lobes (in Tridactylidae, the paraproctal lobes are unmodified and cylindrical in shape) [2,3,4,5,6,7]. Ripipterygids mostly existnear freshwater habitats, such as streams, lakes, ponds, sandy banks, and floodplain areas, where they live mostly in sandy areas [5,8,9].

To date, 249 species of Tridactyloidea have been recorded, including 74 species from three genera of Ripipterygidae [10]. Currently, all extant Ripipterygidae species are distributed in South and Central America. Fossil Ripipterygidae are only represented by two species from ambers, *Mirhipipteryx antillarum* Heads, 2010, from early Miocene (Burdigalian) Dominican amber and *Archaicaripipteryx rotunda* Xu, Zhang, Jarzembowski and Fang, 2020, from mid-Cretaceous Burmese amber [6,11]. Herein, we reviewed the genus of *Magnidactylus* Xu, Fang and Jarzembowski, 2020, and transfered it from Tridactylidae to Ripipterygidae. We also described two new species of Ripipterygidae, *Magnidactylus mirus* sp. nov. and *Magnidactylus gracilis* sp. nov., based on four ambers from northern Myanmar.

## 2. Materials and Methods

The specimens weredeposited at the Department of Plant Protection of Sichuan Agricultural University(SICAU), Chengdu, China (Jun-Jie, Gu, Curator), or in the fossil insect collection at the Key Lab of Insect Evolution and Environmental Changes, Capital Normal University (CNU), Beijing, China (Dong Ren, Curator). In this study, all ambers were collected from the Hukawng Valley, Myitkyina District, Kachin Provincein northern Myanmar [12,13]. The age of the deposits in Tanai Village has been estimated as ca. 99 Ma (98.8 ± 0.6; earliest Cenomanian) based on the UePb dating of zircons from the volcaniclastic matrix of the ambers [14].

The amber containing the specimen was ground and polished on the right size. Photographs were taken with a SZX16 microscope system and cellSens Dimension 3.2 software (Olympus, Tokyo, Japan). In most instances, incident and transmitted light were used simultaneously. All the images weredigitally stacked photomicrographic composites of approximately 20 individual focal planes obtained using Helicon Focus 6 (http://www.heliconsoft.com accessed on 12 May 2022) for a better illustration of the 3D structures.

The morphological terminology ofthe wing venation used heregenerally followed the work by Ragge [15], which was also followed by Azar and Nel [16], with a slight adjustment by Béthoux and Nel [17,18]. Morphological terminology for the body used heregenerally followed the work by Günther [3] and Heads [2]. Wing venation abbreviations were as follows: ScA, anterior Subcosta; ScP, posterior Subcosta; R, Radius; and M, Media. The abbreviations of the characters are as follows: Ocellus, oc; hindwing, hw; tegmen, te; protibia, pt; mesothoracic leg, meso; metatarsus, mt; metafemur, mf; prothoracic leg, pro; apical spur, asp; subapical spur, sasp; epiproct, ep; metatibia, mti; metatarsus, mta; subapical denticular process of metatibia, sadent; cercus, ce; paraproctal lobe, pptl; dorsal valve, dv; and ventral valve, vv.

## 3. Results

### 3.1. Systematic Palaeontology

Orthoptera: Caelifera: Tridactyloidea; Ripipterygidae

Genus *Magnidactylus* Xu, Fang and Jarzembowski, 2020

Emended diagnosis.Protibia slightly inflated distally with four or five dactyls; metatibia lacking swimming plates, metatarsus over twice as long as apical spurs, with a subapical denticular process; paraproctal lobe modified and two-segmented, clavate, longer than the cercus.

Type Species. *Magnidactylus robustus* Xu, Fang and Jarzembowski, 2020

#### 3.1.1. *Magnidactylus mirus* sp. nov. Gu, Zheng, Cao et Yue

Material. Holotype, CNU-ORT-MA2016021, female; a nearly complete specimen of an adult, including the head, prothoracic legs, mesothoracic legs and terminal abdomen, which were preserved well. Metatibiae were obscured by metafemurs and the tegmina overlapped on hindwings. Paratypes, CNU-ORT-MA2018012, female and 6-1014, sex unknown, abdominal crushing, left metatarsustruncated by the edge of the amber, and right metatarsus partially obscured by the bubble.

Diagnosis. Posterior margin of the compound eye was prominent, close to the edge of pronotum; the base of tegmen, mesofemur and metafemur with light coloration; metatibia with relatively small spines on the dorsal margins; the basal segment and the apical segment of the paraproctal lobe was equally thick and clavate.

Etymology. The specific epithet is from the Latin ‘mirus’, which is used to describe theunique shape of compound eyes.

Locality and horizon. Hukawng Valley, Kachin Province, Myanmar; lowermost Cenomanian, Upper Cretaceous.

Description. Holotype, CNU-ORT-MA2016021 (Figure 1), Paratypes, 6-1014 (Figure 2A,B), CNU-ORT-MA2018012 (Figure 2C–F). The measurements based on these three specimens are as follows: Body length 3.03 ± 0.3 mm long (measured from the head to the abdominal apex); the head was 0.44 ± 0.07 mm long; the tegmina length was 1.05 ± 0.1 mm; the hind wing length was 2.39 ± 0.25 mm; the profemur was 0.35 ± 0.04 mm; the protibia was 0.26 ± 0.1 mm; the mesofemur was 0.99 ± 0.6 mm; the mesotibia was 0.72 ± 0.04 mm; the metafemur was 1.75 ± 0.25 mm; the metatibia was 1.62 ± 0.26 mm; the metatarsus was 0.38 ± 0.04 mm; the apical metatibial spur was 0.17 ± 0.01 mm; the subapical metatibial spur was 0.09 ± 0.01 mm; the pronotum was 0.57 ± 0.07 mm long; the cercus was 0.26 ± 0.01 mm; and the paraproctal lobe was 0.35 ± 0.04 mm.

The head dark brown, triangular in shape hypognathous; vertex somewhat inflated forward, without setae; frons slightly convex; face smooth and broad; compound eye large, not typically oval shaped, the posterior margin of the compound eye prominent, close to the edge of the pronotum (Figure 1A,B and Figure 2A–D); ocellus present and tiny; an approximately interocular distance half of the width of the compound eye; moniliform antennae nine segmented (eight segments preserved in the holotype) and inserted beneath the lower margin of compound eye, the flagellomere covered with short setae, scape robust, similar to flagellomere in shape, and flagellomere widening towards apex.

The pronotum large, nearly shield like in dorsal view, dark brown and smooth; lateral margins with a row of short setae; posterior margin broadly rounded; precoxal bridge of the prosternum well developed and visible laterally.

The tegmen dark and sclerotised, base with light coloration; tegmen slightly shorter than metafemur, with four longitudinal veins visible, without covering of setae; ScA slightly curved; ScP simple and nearly straight; M nearly straight and parallel to ScP; R slightly curved and fuse with M at apex. Hindwing brown with a transparent area light at the apical part, over twice as long as tegmen, surpassing terminal abdomen, longer than metafemur and without setae.

The legs: Prothoracic leg—prothoracic leg brown, markedly shorter than mesothoracic leg; profemur narrow from base to apex, with long dorsal and ventral setae; protibia robust, slightly inflated distally with a dense covering of setae and four dactyls; protarsus two-segmented, slender, with the second segment longer than the first; basitarsus short; apical tarsomere elongate, slightly curved; pretarsus with two claws. Mesothoracic leg—mesofemur brown, with light coloration near the middle and sparse setae on the ventral margins; mesofemur slender, curved, over twice as long as the prefemur, basally narrow and apically broad; mesotibia brown, apex and near base with light coloration; mesotibia approximately twice as long as the protibia, with dorsal and ventral setae; mesotarsus two-segmented, slightly longer than the protarsus; basitarsus short, expanding into globular structure with two protuberances; pretarsus with two long claws. Metathoracic leg: saltatorial; metafemur brown, with light coloration at its base and apex; metafemur slightly shorter than abdomen and greatly inflated along its entire length, with prominent dorsal carina; genicular lobe large and well developed; metatibia brown, with light coloration near the base; metatibia slightly shorter than metafemur, very slender, quadrate in section, with tiny spines on the dorsal margins, without setae; metatibia lacking swimming plates, with two apical spurs and two subapical spurs, subapical spurs shorter than apical spurs; metatarsus one-segmented, slender, broader in the middle, over twice as long as apical spurs, with a tiny subapical denticular process (Figure 2E,F).

The abdomen dark brown, with sparse and long setae; cercus unsegmented, cylindrical, with few long and thin setae; paraproctal lobe, longer than cercus, two-segmented, the basal segment and the apical segment of paraproctal lobe equally thick and clavate, with compact ventral setae, long and thick; dorsal valve curves upward, slightly shorter than paraproctal lobe, each dorsal valve with a elongated hook; ventral valve shorter than dorsal valve, each ventral valve with a powerful side tooth at the apex (Figure 1C,D).

#### 3.1.2. *Magnidactylus*
*gracilis* sp. nov. Gu, Zheng, Cao et Yue

Material. Holotype, SICAU-A-085, female, was a nearly complete specimen of an adult. The prothoracic legs, pronotum, tegmina and abdomen were partially obscured by the bubble. Head, mesothoracic legs, metathoracic legs and paraproctal lobes were well preserved.

Diagnosis. Metatibia with relatively large spines on the dorsal margins; metatarsus with a large subapical denticular process forming a fork with the apical denticular process; apical segment of paraproctal lobe distinctly swollen.

Etymology: The specific epithet is from the Latin ‘gracilis’ and is used to describethe relatively slender and straight mesotibia in the *Magnidactylus*.

Locality and horizon. Hukawng Valley, Kachin Province, Myanmar; lowermost Cenomanian, Upper Cretaceous.

Description. Holotype, SICAU–A–085 (Figure 3A,B and Figure 4A–F). The measurements were as follows: 5.92 mm long (measured from the head to the abdominal apex); the head was 1.24 mm long; the tegmen length was 1.5 mm; the hindwing length was 4.04 mm; the profemur was 0.6 mm; the protibia was 0.58 mm; the mesofemur was 1.43 mm; the mesotibia was 1.17 mm; the metafemur was 2.87 mm; the metatibia was 2.36 mm; the metatarsus was 0.58 mm; the apical metatibial spur was 0.19 mm; the subapical metatibial spur was 0.17 mm; the pronotum was 1.08 mm long; the cercus was 0.37 mm; and the paraproctal lobe was 0.59 mm.

The head dark brown, triangular in shape hypognathous; vertex somewhat inflated forward, with sparse short setae; head somewhat laterally compressed, frons observably convex; face broad; compound eye large and suboval; ocellus present, relatively large; due to head compression, interocular distance uncertain; moniliform antennae nine segmented and inserted beneath the lower margin of compound eye, the flagellomere covered with short setae, scape robust, similar to flagellomere in shape, flagellomere widening towards apex.

The pronotum large, nearly shield like in dorsal view, dark brown, with sparse short setae; posterior margin broadly rounded; precoxal bridge of prosternum well developed and visible laterally.

The tegmen dark and sclerotised, slightly shorter than metafemur; tegmen obscured by bubbles and metathoracic leg, only two longitudinal veins visible, without covering of the setae; ScP slightly curved; M simple and nearly straight. Hindwing dark brown, over twice as long as the tegmen, surpassing terminal abdominal, longer than metafemur and without setae.

The legs: Prothoracic leg—prothoracic leg brown, markedly shorter than mesothoracic leg; profemur squeezed out of shape, with long dorsal and ventral setae; protibia robust, slightly inflated distally with a dense covering of setae and four strong dactyls; protarsus two-segmented, slender, with second segment longer than first; basitarsus short; apical tarsomere elongate, slightly curved; pretarsus with two slender claws. Mesothoracic leg—mesofemur dark brown, with light coloration near the middle and apex, sparse setae on the dorsal margins; mesofemur slender, curved, approximately 1.5 times as long as profemur, broader in the middle; mesotibia brown, with light colorationat apex and near the base; mesotibia approximately twice as long as protibia, slender and straight, uninflated, with dorsal and ventral setae; mesotarsus two-segmented, slightly longer than protarsus; basitarsus short, expand into globular structure with two protuberances; pretarsus with two long claws. Metathoracic leg—saltatorial; metafemur dark brown, with light coloration at base; metafemur slightly shorter than abdomen and greatly inflated along its entire length, with prominent dorsal carina; genicular lobe large and well developed; metatibia brown, with light coloration near the base; metatibia slightly shorter than metafemur, very slender, quadrate in section, with relatively large spines on the dorsal margins, without setae; metatibia lacking swimming plates, with two apical spurs and two subapical spurs, subapical spurs shorter than apical spurs; metatarsus one-segmented, over twice as long as apical spurs, forked apically, with a large and obvious subapical denticular process, ventral margins with a row of dense marginal setae.

The abdomen dark brown, with sparse long setae; cerci unsegmented, cylindrical, with few long and fine setae; paraproctal lobe clavate, longer than cercus, two-segmented, apical segment distinctly swollen, with compact setae, long and thick; epiproct triangular, longer; dorsal valve curves upward, shorter than paraproctal lobe, dorsal margins with irregularly denticles; ventral valve shorter thandorsal valve, with long setae, each ventral valvewith a powerful side tooth at apex (Figure 4E,F).

### 3.2. Key to Species for Magnidactylus

1 Apical segment of paraproctal lobe distinctly swollen in female….***M. gracilis* sp. nov.**

– Apical segment of paraproctal lobe no swollen in female……………………**2**

2 Basal segment and apical segment of paraproctal lobe equally thick and clavate in female……………………………………………………***M. mirus***
**sp. nov.**

– Basal segment of paraproctal lobe swollen and obviously thicker than apical segment in female…………………………***M. robustus* Xu, Fang and Jarzembowski, 2020**

## 4. Discussion

The genus *Magnidactylus* was described by Xu et al. [11] and assigned to Tridactylidae with the following characteristics: Presence of a precoxal bridge connecting the prosternum and the pronotum; pro- and meso-tarsi with two tarsomeres, metatarsi with a single tarsomere; and cerci, which was two-segmented [11]. However, most of the characters mentioned above are general characters in Tridactyloidea, which cannot distinguish Tridactylidae from Ripipterygidae [19,20]. It is worth noting that a pair of terminal abdominal structures of *M. robustus* was interpreted as cerci by the authors. However, based on the illustrations and photos of the specimen (Figure 1C,D; Figure 4E,F; from Reference [11]), this structure is born on the paraproct, and should be interpret as paraproctal lobes. In female Ripipterygidae, the paraproctal lobes are always two-segmented and the apical segments are laterally flattened [3]. Therefore, the holotype of *M. robustus* should be a female. From the detailed photos and illustration of the end of the abdomen of *M. robustus* (Figure 4E,F; from Reference [11]), a cylindrical and unsegmented structure located aside the paraproctal lobes should be its cerci. This pair of unsegmented cerci indicates that *Magnidactylus* cannot be assigned to any subfamily of Tridactylidae, except the Mongoloxyinae. *Birmitoxya intermedia* Gorochov (2010) is the only species of Mongoloxyinae showing complete abdominal structures [21,22,23]. Compared to *B. intermedia*, *M. robustus* has developed paraproctal lobes longer than cerci. On the contrary, *M. robustus* and the two new species described here share with Ripipterygidae pro- and meso-tarsi with only two tarsomeres, metatarsi with only one tarsomere, unsegmented cerci and developed paraproctal lobes. Thus, the genus *Magnidactylus* should be assigned to Ripipterygidae rather than to Tridactylidae.

Ripipterygidae consist of only one subfamily with two extant genera, and an extinct genus *Archaicaripipteryx* (Xu, Zhang, Jarzembowski and Fang, 2020), without subfamily assignment. *Magnidactylus* is similar to *Archaicaripipteryx* with the following characteristics: Body dark, with several light patches; metatarsi over twice as long as the apical metatibial spurs, with a subapical denticular process; and a cylindrical cerci. Compared to *Archaicaripipteryx*, *Magnidactylus* has modified, two-segmented and clavate paraproctal lobes, which are longer than cerci(in *Archaicaripipteryx*, paraproctal lobes cylindrical, about the same length as cerci, unsegmented; Xu et al. (2020) mistakenly interpreted this structure as cerci) [24]. *Ripipteryx* Newman (1834) is the most diverse genus of Ripipterygidae, *Magnidactylus* shares a dark body with several light patches (in *Ripipteryx*, colorationis generally black or very dark brown, most often with starkly contrasting white, yellow and occasionally red markings forming distinctive maculae); cerci cylindrical; the female with modified paraproctal lobes, two-segmented, longer than cerci (in *Ripipteryx*, the female with two-segmented paraproctal lobes, apical segment are laterally flattened). But, *Magnidactylus* differs from *Ripipteryx* in the following characters: metatarsi over twice as long as the apical metatibial spurs (in *Ripipteryx*, the apical metatibial spurs are usually equal in length or only slightly longer than the metatarsi); the presence of a distinctive subapical denticular process on the metatarsi (in *Ripipteryx*, metatarsus without subapical denticular process) [7,25,26]. *Magnidactylus* shares with *Mirhipipteryx* Günther, 1969 modified, two-segmented paraproctal lobes in female. *Magnidactylus* differs from *Mirhipipteryx* with the following characteristics: Dark body with several light patches (*Mirhipipteryx* species are usually dark brown or black, but lack the light patches); metatarsi is over twice as long as the apical metatibial spurs, with a subapical denticular process (in *Mirhipipteryx*, the apical metatibial spurs at least twice as long as the metatarsi) [5,6,27]. Therefore, *Magnidactylus* can be separated from all known genera of Ripipterygidae.

The two new species described here can be assigned to *Magnidactylus* by the metatarsi that is over twice as long as the apical metatibial spurs, with a subapical denticular process; modified and two segmented paraproctal lobes. Only females were recorded for all three species. *M. robustus* differs from the two new species withthe following characteristics: Basal segment of paraproctal lobes are swollen and obviously thicker than apical segment (in *M. mirus*, the basal segment and the apical segment of paraproctal lobes are equally thick and clavate; in *M. gracilis*, apical segment of paraproctal lobes are distinctly swollen); the metatarsi are straight and with a tiny subapical denticular process (in *M. mirus*, the metatarsi is broader in the middle; in *M. gracilis*, the metatarsi with a large subapical denticular process forming a fork with apical denticular process); *M. robustus* body size is larger than *M. mirus* and similar to *M. gracilis* (*M. robustus* body 6.3 mm long measured from head to abdominal apex; *M. gracilis* is about 5.92 mm long; and *M. mirus* is only 3.03 ± 0.3 mm long); protibiae with five dactyls (in *M. mirus* and *M. gracilis*, protibia with four dactyls); the compound eyes are typically ovoid (similar with *M. gracilis* but differ from *M. mirus*, in *M. mirus*, the posterior margin of compound eyes prominent, close to the edge of pronotum). Taken together, this suggests that *M. mirus* and *M. gracilis* can be established as two separate species and placed in *Magnidactylus*.

Owing to the poor preservation of compression fossil tridactyloids, they are hard to compare with amber and extant species. Thus, we propose a key to genera for amber tridactyloids [6,11,16,20,23,28,29,30]. Since *Cascogryllus* Poinar (2020) work did not include a subapical denticular process on the metatarsi, we considered that it belonged to Tridactylinae [4,28]. Among all genera covered in the key, only *Birmitoxya* Gorochov (2010) is based on larvae (male middle or late instar nymph) [21], which are distinguished from adults mainly by the presence or absence of fully developed wings, and characters used in the key can be observed in this genus.

### Key to Genera for Tridactyloidea in Amber

1 Mesotibia uninflated, cercus unsegment…………**2. Ripipterygidae Ander, 1939**

– Mesotibia distinctly inflated, cercus two-segment………**4. Tridactylidae Brullé, 1835**

2 The apical metatibial spurs longer than metatarsus……***Mirhipipteryx* Günther, 1969**

– The apical metatibial spurs shorter than metatarsus or of equal length to metatarsus……………………………………………………………………**3**

3 Paraproctal lobe normal, cylindrical, unsegmented in female………………………………………***Archaicaripipteryx* Xu, Zhang, Jarzembowski and Fang, 2020**

– Paraproctal lobe modified, longer than cercus, two-segmented in female…………………………………………***Magnidactylus* Xu, Fang and Jarzembowski, 2020**

4 Paraproctal lobe absent or underdeveloped, shorter than the genital plate……………………………….***Birmitoxya* Gorochov, 2010 (MongoloxyinaeGorochov, 1992)**

– Paraproctal lobe well-developed, longer than the genital plate…………………**5**

5 Metatarsus with subapical denticular process…….**6. Dentridactylinae Günther, 1979**

– Metatarsus without subapical denticular process………..**8. Tridactylinae Brullé, 1835**

6 Metatibia with swimming plates……………***Guntheridactylus* Azar and Nel, 2008**

– Metatibia without swimming plates………………………………………**7**

7 Hindwing absent……………………………***Burmadactylus* Heads, 2009**

– Hindwing present…………………………***Paraxya* Cao, Chen and Yin, 2019**

8 Prosternum with a pair of erect multicellular tubercles…….***Cascogryllus* Poinar, 2020**

– Prosternum without erect multicellular tubercles……………………………**9**

9 Metatarsus vestigial, reduced to a tiny nub………………***Ellipes* Scudder, 1902**

– Metatarsus well developed…………………………………………**10**

10 Wing absent……………………***Phyllotridactylus* Xu, Wang, Fan et al., 2021**

– Wing present……………………………………………………**11**

11 Metatibia with two rows of denticles, and the apical metatibial spurs shorter than metatarsus………………………***Amberotridactylus* Du, Xu and Zhang, 2022**

– Metatibia without denticles, and the apical metatibial spurs longer than metatarsus…………………………………………………***Archaeoellipes* Heads, 2010**

## 5. Conclusions

Based on the morphological analysis above, we revised *Magnidactylus* Xu, Fang and Jarzembowski 2020 and transferred it from Tridactylidae to Ripipterygidae. *Magnidactylus mirus* sp. nov. and *Magnidactylus gracilis* sp. nov. were erected based on four amber specimens from the northern Myanmar. These new findings increase the diversity of fossil Tridactyloidea and provide new morphological knowledge of Ripipterygidae.

## Figures and Tables

**Figure 1 insects-13-00979-f001:**
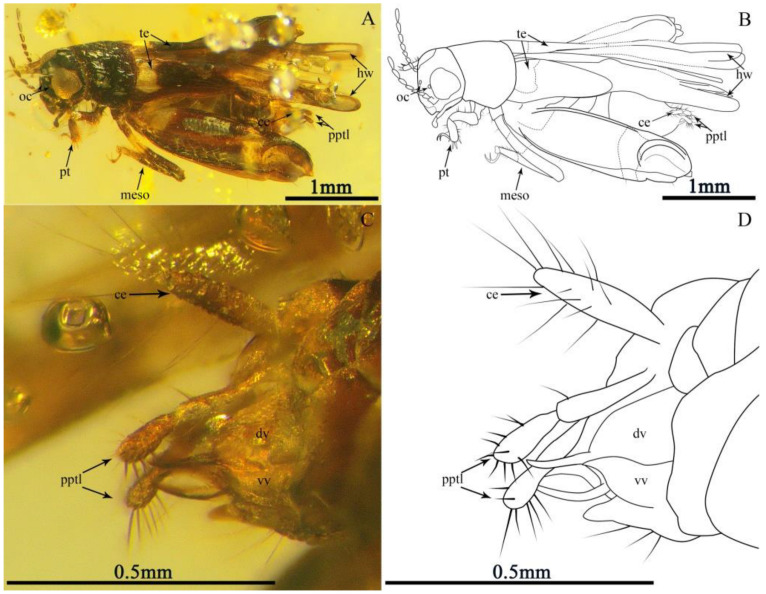
Holotype (CNU-ORT-MA2016021) of *Magnidactylus*
*mirus* sp. nov., holotype; (**A**) photograph of the habitus in the dorsal-lateral view, (**B**) line drawing of the habitus, (**C**) photograph of the terminal abdomen, and (**D**) line drawing of the terminal abdomen.

**Figure 2 insects-13-00979-f002:**
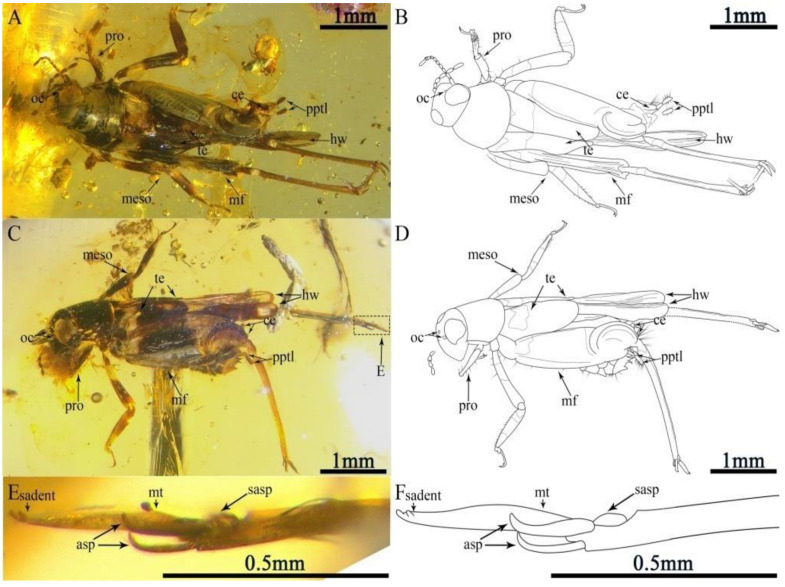
Paratypes of *Magnidactylus mirus* sp. nov. (**A**,**B**) Habitus and linedrawing of 6-1014 in dorsal view; (**C**,**D**) habitus and linedrawing of CNU-ORT-MA2018012 in the lateral view, and (**E**,**F**) photograph and linedrawing of the right metatarsus (CNU-ORT-MA2018012) (photograph obtained from the other side of specimen CNU-ORT-MA2018012).

**Figure 3 insects-13-00979-f003:**
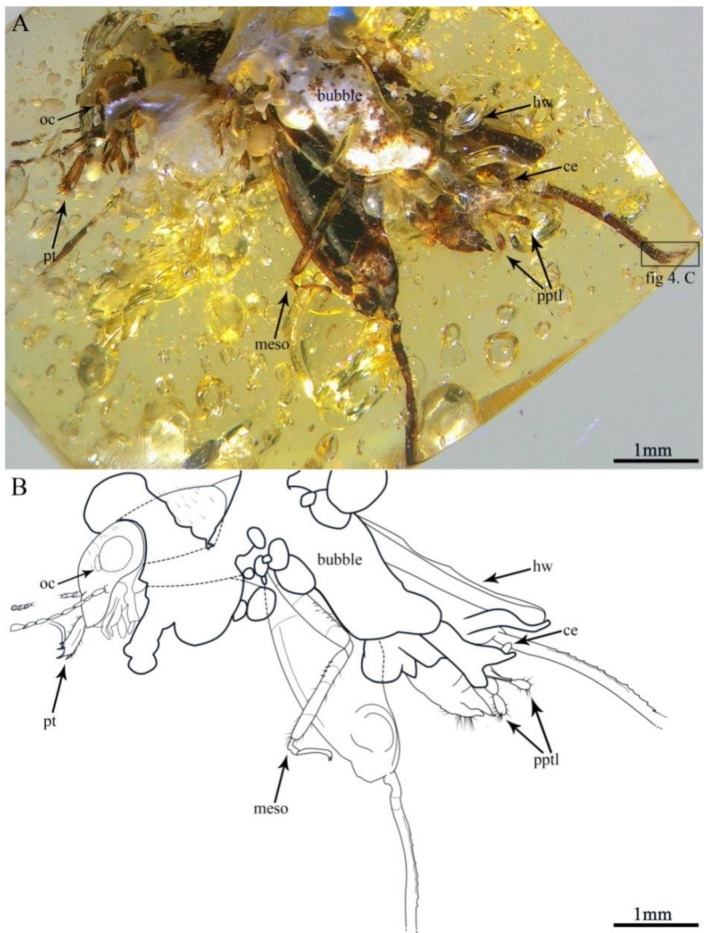
Habitus of *Magnidactylus gracilis* sp. nov., holotype (SICAU–A–085). (**A**) Photograph in lateral view and (**B**) line drawing in lateral view.

**Figure 4 insects-13-00979-f004:**
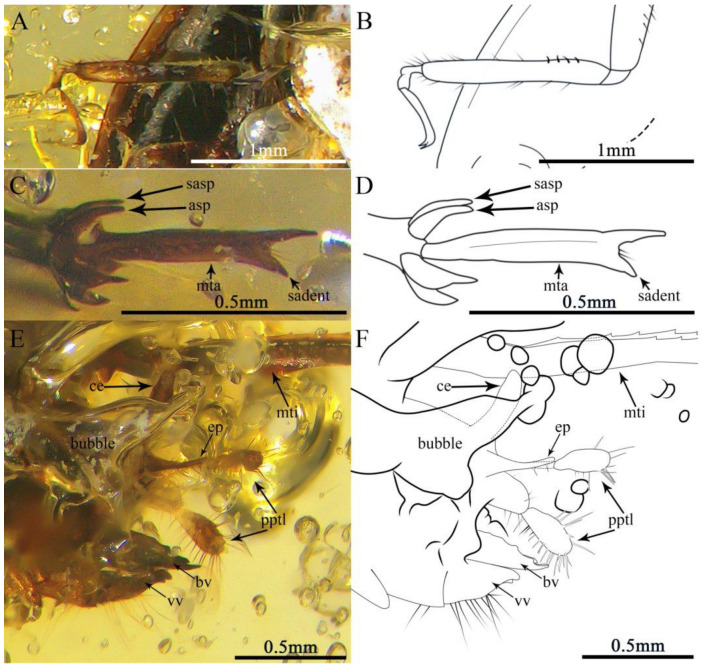
Details of *Magnidactylusgracilis* sp. nov.: (**A**) photograph of left mesotibia, (**B**) line drawing of left mesotibia, (**C**) photograph of right metatarsus, (photograph obtained by holding specimen SICAU–A–085 upright with its side up), (**D**) line drawing of right metatarsus, (**E**) photograph of terminal abdominal and (**F**) line drawing of terminal abdominal.

## Data Availability

No new data were created or analyzed in this study. Data sharing isnot applicable to this article.

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
