# Peer review of "Two New Species of Ripipterygidae (Orthoptera, Tridactyloidea) from Mid-Cretaceous of Myanmar with a Key to the Genera of Tridactyloidea in Amberâ€"

_insects, 2022, doi:10.3390/insects13110979_

Round 1
Reviewer 1 Report
This is an interesting manuscript that can be published. Two serious remarks.
1. Give a key to species of the genus Magnidactylus from Burmese amber.
2. Give the authors of the genera and designate the ambers where these genera were found for the genera key.
3. Some minor flaws that I corrected in the text.

Author Response
Thank you very much for your kind and valuable revisions, we have modified and corrected all the issues.
- Give a key to species of the genus Magnidactylus from Burmese amber.
Reply: we added a key to species of Magnidactylus in text
- Give the authors of the genera and designate the ambers where these genera were found for the genera key.
Reply: we added the authors of the genera
- Some minor flaws that I corrected in the text.
Reply: we corrected the flaws.
Reviewer 2 Report
Title must be corrected as follow:
Two new species of Ripipterygidae (Orthoptera, Tridactyloidea) from mid-Cretaceous Myanmar with a key to genera of Tridactyloidea in amber
Page 2 line 68
Printed: [2].Wing
Replace to: [2]. Wing
Page 2 lines 72 and 73
Printed: matetibia
Replace to: metatibia
Page 2 line 82
Printed: Magnidactylus mirus sp. nov. Gu, Zheng, Cao et Yue
Replace to: Magnidactylus mirus Gu, Zheng, Cao et Yue, sp. nov.
Page 2 lines 83-87
Printed:
Material. Holotype, CNU-ORT-MA2016021, female. nearly complete specimen of an adult: head, prothoracic legs, mesothoracic legs and terminal abdomen preserved well; matetibiae obscured by metafemurs; tegmina overlapped on hindwings. Paratype, 85 CNU-ORT-MA2018012, female; 6-1014, sex unknown, abdominal crushing, left metatarsus truncated by the edge of the amber, right metatarsus is partially obscured by the bubble.
Replace to: Material. Holotype, CNU-ORT-MA2016021, female; nearly complete specimen of an adult: head, prothoracic legs, mesothoracic legs and terminal abdomen preserved well; metatibiae obscured by metafemurs; tegmina overlapped on hindwings. Paratypes, 85 CNU-ORT-MA2018012, female; and 6-1014, sex unknown, abdominal crushing, left metatarsus truncated by the edge of the amber, right metatarsus is partially obscured by the bubble.
Page 3 lines 109-110
Printed: Figure 1. Holotype (CNU-ORT-MA2016021) of Magnidactylus mirus sp. nov., holotype : (A) 109 photograph of habitus in dorsal view, (B) line drawing of habitus, ....
Replace to: Figure 1. Holotype (CNU-ORT-MA2016021) of Magnidactylus mirus sp. nov., holotype: (A) photograph of habitus in dorso-lateral view, (B) line drawing of habitus, ....
Page 3 line 117
Printed: preserved in the holotype)and inserted beneath
Replace to: preserved in the holotype) and inserted beneath
Page 3 line 119
Printed: about 1.5 times as long as flagellomere, pedicel much
Replace to: about 1.5 times as long as pedicel, pedicel much
Page 4 line 154
Printed: Figure 2. Paratypes of Magnidactylus mirus sp. nov.. A, B: habitus and linedrawing of 6-1014 in
Replace to: Figure 2. Paratypes of Magnidactylus mirus sp. nov.. A, B: habitus and linedrawing of 6-1014 in
Page 5 line 158
Printed: Magnidactylus gracilis sp. nov. Gu, Zheng, Cao et Yue
Replace to: Magnidactylus gracilis Gu, Zheng, Cao et Yue, sp. nov.
Page 5 line 162
Printed: Diagnosis. matetibia with relatively large spines on the dorsal margins;
Replace to: Diagnosis. Metatibia with relatively large spines on the dorsal margins;
Page 5 line 166
Printed: slender and straight mesotibia in the Magnidactylus.
Replace to: slender and straight mesotibia in the Magnidactylus.
Page 5 line 178
Printed: Figure 3. Habitus of Magnidactylus gracilis sp. nov., holotype (SICAU–A–085): (A) photograph
Replace to: Habitus of Magnidactylus gracilis sp. nov., holotype (SICAU–A–085): (A) photograph
Page 7 lines 227-228
Printed: The genus Magnidactylus Xu, Fang & Jarzembowski 2020 was erected by Xu et al. [11], and assigned to Tridactylidae by the following characters:
Replace to: The genus Magnidactylus was erected by Xu et al. [11], and assigned to Tridactylidae by the following characters:
Page 7 lines 245-246
Printed: Thus, Magnidactylus Xu, Fang & Jarzembowski, 2020 should be assigend to Ripipterygidae rather than to Tridactylidae.
Replace to: Thus, the genus Magnidactylus should be assigned to Ripipterygidae rather than to Tridactylidae.
Page 8 line 278
Printed: denticular process ( in M. mirus, the metatarsi
Replace to: denticular process (in M. mirus, the metatarsi
Page 9 (key), paragraph 9
Printed: 9 Metatarsus vestigial, reduced to a tiny nub nestled …………………… Ellipes
Replace to: 9 Metatarsus vestigial, reduced to a tiny nub …………………… Ellipes
Page 9 line 297
Printed: fossil Tridactyloidea and provide new knowledge of morphology in mud crickets.
Replace to: fossil Tridactyloidea and provide new knowledge of morphology in pygmy mole crickets.
Page 9 line 318
Printed: (Saltatoria, Insecta), Mitt. Zool. Mus.
Replace to: (Saltatoria, Insecta). Mitt. Zool. Mus.
Page 9 line 319
Printed: Tridactylidae Brunner und zur Klassifi kation der
Replace to: Tridactylidae Brunner und zur Klassifikation der
Page 10 line 330
Printed: Orthoptera Species File. V ersion 5.0/5.0. 2022.
Replace to: Orthoptera Species File. Version 5.0/5.0. 2022.
Page 10 line 344
Printed: Béthoux, O.; Nel, A. V. enation pattern and revision
Replace to: Béthoux, O.; Nel, A. Venation pattern and revision
Page 10, line 351
Printed: Gorochov, A. V.; Jarzembowski, E. A. Coram R A. Grasshoppers
Replace to: Gorochov, A. V.; Jarzembowski, E. A.; Coram, R. A. Grasshoppers
Author Response
Dear reviewer:
Thank you for your kind and detailed revisions, we have accepted most of your corrections.
Jun-Jie

Reviewer 3 Report
This paper presents the descriptions of two new species of pigmy crickets from the Cretaceous; the genus is transferred to a different family and a key is given for fossil genera found in amber.
This paper suffers from many editing errors, which must be corrected by the authors. See attached file.
I am not sure the authors have exploited all the characters visible on the fossils, especially gracilis.
Also, for the key, can all the characters be observed on all the fossils mentioned in the key ? male versus female characters ? juvenile versus adults ? The authors should introduvce the families and if possible the subfamilies into the key.

Author Response
Dear reviewer:
thank you for you kind and valuable comments, we have corrected as your suggestions.
This paper suffers from many editing errors, which must be corrected by the authors. See attached file.
Reply: thank you very much, we have checked again and correct the erros.
I am not sure the authors have exploited all the characters visible on the fossils, especially gracilis.
Reply: we have checked the specimens again and improve the description and modified the linedrawings of Magnidactylus gracilis sp. nov.
Also, for the key, can all the characters be observed on all the fossils mentioned in the key ? male versus female characters ? juvenile versus adults ? The authors should introduvce the families and if possible the subfamilies into the key.
Reply: among all genera covered in the key, only Birmitoxya Gorochov, 2010 is based on larvae (male middle or late instar nymph), which are distinguished from adults mainly by the presence or absence of fully developed wings, and characters used in the key can be observed in this genus. And we modified the key as your suggestion.
Round 2
Reviewer 1 Report
The manuscript has been corrected according to the comments. I recommend accepting this ms.
Author Response
Dear reviewer:
thank you for your kind revision.
Best wishes
Jun-Jie